# Inductive Graph Alignment Prompt: Bridging the Gap between Graph Pre-training and Inductive Fine-tuning From Spectral Perspective.

## ABSTRACT

The "Graph pre-training and fine-tuning" paradigm has significantly improved Graph Neural Networks(GNNs) by capturing general knowledge without manual annotations for downstream tasks. However, due to the immense gap of data and tasks between the pre-training and fine-tuning stages, the model performance on downstream task is still limited. Inspired by prompt fine-tuning in Natural Language Processing(NLP), many endeavors have been made to bridge the gap in graph domain. But existing methods simply reformulate the form of fine-tuning tasks to the pre-training ones, ignoring the inherent gap of graph data. With the premise that the pre-training graphs are compatible with the fine-tuning ones, these methods typically operate in transductive setting. In order to generalize graph pre-training to inductive scenario where the fine-tuning graphs might significantly differ from pre-training ones, we propose a novel graph prompt based method called Inductive Graph Alignment Prompt(IGAP). Firstly, we unify the mainstream graph pre-training frameworks and analyze the essence of graph pre-training from graph spectral theory. Then we identify the two sources of the data gap in inductive setting: (i) graph signal gap and (ii) graph structure gap. Based on the insight of graph pre-training, we propose to bridge the graph signal gap and the graph structure gap with learnable prompts in the spectral space. A theoretical analysis ensures the effectiveness of our method. At last, we conduct extensive experiments among nodes classification and graph classification tasks under the transductive, semi-inductive and inductive settings. The results demonstrate that our proposed method can successfully bridge the data gap under different settings.

## KEYWORDS

Graph Pre-training, Graph Prompt, Graph Neural Networks

ACM Reference Format:

Anonymous Author(s). 2018. Inductive Graph Alignment Prompt: Bridging the Gap between Graph Pre-training and Inductive Fine-tuning From Spectral Perspective.. In *Proceedings of Make sure to enter the correct conference title from your rights confirmation emai (Conference acronym 'XX)*. ACM, New York, NY, USA, 13 pages. https://doi.org/XXXXXXX.XXXXXXX

## 1 INTRODUCTION

Graph Neural Networks(GNNs) taking advantage of message passing to fuse node features and topological structures, have been successfully applied in various applications such as Web search, personal recommendation and community discovery [10, 12, 24, 35, 43]. Traditional GNNs are trained under a supervised manner which not only necessitates laborious manual annotations but also is susceptible to over-fitting problem. Inspired by the success of the pre-training model in Natural Language Processing(NLP) [3, 7, 8, 21, 23, 30] and Computer Vision(CV) [5, 6, 11, 15, 22, 29], many endeavors have been paid into transplanting the ethos of "pre-training and fine-tuning" into the domain of graph.

**Graph Pre-training and fine-tuning.** This paradigm involves two distinct stages: (i) graph pre-training stage and (ii) fine-tuning stage. During the graph pre-training stage, GNNs glean general patterns from unannotated data, encompassing many intrinsic graph properties such as local node feature distributions, topological patterns and the consistent fusion of local and global graph patterns. Subsequently, in the fine-tuning stage, the GNNs initialized with the pre-trained parameters can be adapted seamlessly to many downstream tasks even with scant labels and training epochs.

Although the paradigm of "graph pre-training and fine-tuning" emancipates GNNs from the burdensome need for extensive manual annotations and empowers them to perceive general graph patterns to improve downstream tasks, there is still an immense gap between the pre-training stage and fine-tuning stage limiting the performance of the pre-trained models. In the domain of NLP, many innovative prompt based methods are proposed to bridge this gap [3, 16, 20–22, 38], whose philosophy lies in reformulating fine-tuning tasks to mirror the format of pre-training objectives thus the pre-trained knowledge can be transferred seamlessly. Similar strategies have been applied in the realm of graphs to narrow down the gap [9, 25, 33, 34]. However, compared to the gap in NLP, it is far more challenging in graph scenario [28], especially under the inductive setting. The inductive setting, where fine-tuning datasets significantly differ from their pre-training counterparts, is prevalent in the application of pre-training models. In NLP, different language datasets are naturally compatible with each other because the semantic information is consistent among them. But in the graph domain, not only the node/edge features might have disparate distributions but also the topological structures differ significantly. This diversity of graph data will result in incompatible patterns between the pre-training and fine-tuning graphs. We give an example to illustrate, considering the NLP domain with distinct corpus of pre-training and fine-tuning datasets: "I feel happy in passing the exam." and "It's happy to win that game.", the token "happy" retains the consistent semantic meaning among them thus can be transferred easily. While in the graph realm, the users from

different communities might have different social patterns even thought they are in the same social network, the knowledge specific to one graph is hard to be transferred to another. Additionally, the pre-training process usually operates as a black box where the pre-training dataset is unavailable. These distinctive traits inherent to inductive scenario raise new challenges for existing methods:

- **Transductive Limitation.** Although many graph prompt based methods have been proposed to bridge the gap caused by pre-training and fine-tuning task types as the language prompts do, these methods still ignore the diversity of graph data [25, 33]. Existing graph prompt based methods operate under the assumption of compatibility between pre-training and fine-tuning graphs, meaning all these methods are all **transductive** where **the performance can only be ensured when GNNs are pre-trained and fine-tuned on the same graph.** Under the inductive setting, the pre-trained GNNs might have sub-optimal performance and even negative transfer on the fine-tuning graphs.
- **Inaccessibility of Pre-training Data.** Due to data privacy concerns, the GNNs' pre-training process often operates as a black box, meaning that we can only get the pre-trained model with the pre-training dataset unavailable. This lack of access complicates the fine-tuning process under the inductive setting. Traditional transfer learning based methods, which require additional information about the pre-training dataset to align the representation of the GNNs, are not suitable for inductive fine-tuning. Graph prompt based methods might also have the compromised performance in the absence of pre-training data as a prompt.

In order to generalize the paradigm of "graph pre-training and fine-tuning" to inductive setting where the fine-tuning graphs significantly differ from their pre-training counterparts, we propose a novel graph prompt based method named Inductive Graph Alignment Prompt(IGAP). To address the the data gap without direct access to the pre-training graph, we first delve into the process of graph pre-training and then design graph prompts according to the characteristics of pre-training for inductive fine-tuning. Specifically, in the graph pre-training stage, we analyze the essence of this process using spectral graph theory. Our key insight reveals that **graph pre-training predominantly aligns the graph signal with low-frequent components rather than high-frequent ones.** Then for the inductive fine-tuning stage, we identify two primary sources contributing to the data gap: (i) graph signal gap and (ii) topological structure gap. These kinds of gap manifest as node features perturbation and spectral space misalignment from graph spectral theory. Based on the understanding of graph pre-training, we propose an innovative solution for inductive fine-tuning. To counteract the influence of graph signal perturbation, we introduce a learnable graph signal prompt that offers adaptive compensations. Additionally, a spectral space alignment prompt is introduced to align the $K$-smallest frequent components, which rectifies spectral space misalignment and makes the transfer of essential knowledge possible. We provide a theoretical analysis to ensure the effectiveness of our method. Finally, we utilize a label prompt to reformulate the fine-tuning task to harmonize with the pre-training objective.

We validate the effectiveness of IGAP through extensive experiments under transductive, semi-inductive, and inductive settings for both node and graph classification tasks. The experimental results demonstrate the better performance of IGAP in bridging the gap between graph pre-training and inductive fine-tuning.

## 2 RELATED WORK

**Graph Pre-training.** Graph pre-training aims at leveraging vast amounts of label-free graph data to equip GNNs with universal graph knowledge. There are three mainstream graph pre-training frameworks: (i) subgraph contrastive based methods such as GraphCL [42], GRACE [45], and GCA [46] train the GNNs by differentiating the negative subgraph patterns from positive ones; (ii) link prediction based methods such as GPT-GNN [14] train the GNNs through the masked link prediction task; (iii) local-global contrastive based methods such as DGI [36], ST-DGI [27] utilize the mutual information to encode global patterns into local representations.

**Graph Transfer Learning.** Graph transfer learning aims at facilitating the knowledge transfer learned from one task to another. It can narrow down the gap between the source and target tasks [26, 31, 39, 44]. These kinds of methods achieve this goal by aligning the distribution of the two datasets with the regularization or generative constraints.

**Prompt and Graph Prompt.** To bridge the gap between pre-training and fine-tuning objectives, many prompt based methods are proposed. Most of the methods are from CV and NLP domains [3, 8, 11, 15, 21, 22, 30], whose common idea lies at reformulating the fine-tuning tasks into the pre-training paradigms. Inspired by this idea, several graph prompt based methods are also proposed: GPPT [33] incorporates learnable graph label prompts, transforming node classification into a link-prediction task to narrow down the task type gap. GraphPrompt [25] unifies the graph prompt templates and enhance their performance through learnable readout prompt functions. All-in-One [34] introduces a novel graph token structure accompanied by a token insertion technique.

However, all these methods are not fit for the inductive fine-tuning because the transfer learning based methods need addition information about the pre-training datasets which is unavailable; The graph pre-training and graph prompt based methods are transductive with the assumption that the pre-training and fine-tuning graphs have compatible patterns.

## 3 PRELIMINARY

In this section we present the preliminary knowledge used in this paper.

**Graph and Graph Laplacian.** Graph $\mathcal{G}$ is denoted as $\mathcal{G} = (\mathcal{V}, \mathcal{E}, X)$, where $\mathcal{V}$ is the set of nodes, $\mathcal{E}$ is the set of edges and $X$ is the graph signal matrix[1]. A set containing $N$ graphs is denoted as $\mathcal{S}_{\mathcal{G}} = \{\mathcal{G}_1, ..., \mathcal{G}_N\}$. The Graph $\mathcal{G}$ can also be represented as $\mathcal{G} = (A, X)$ where $A$ represents the adjacent matrix. The graph Laplacian $L$ of $\mathcal{G} = (A, X)$ is calculated as $L = D - A$ where $D$ is the degree matrix. The graph Laplacian $L$ is a real symmetric matrix thus can be diagonalized as [4]:

$$L = U \Lambda U^T \qquad (1)$$

---

[1]This can also be called node features, for the convenience of graph signal process, we call it node signal matrix in this paper.

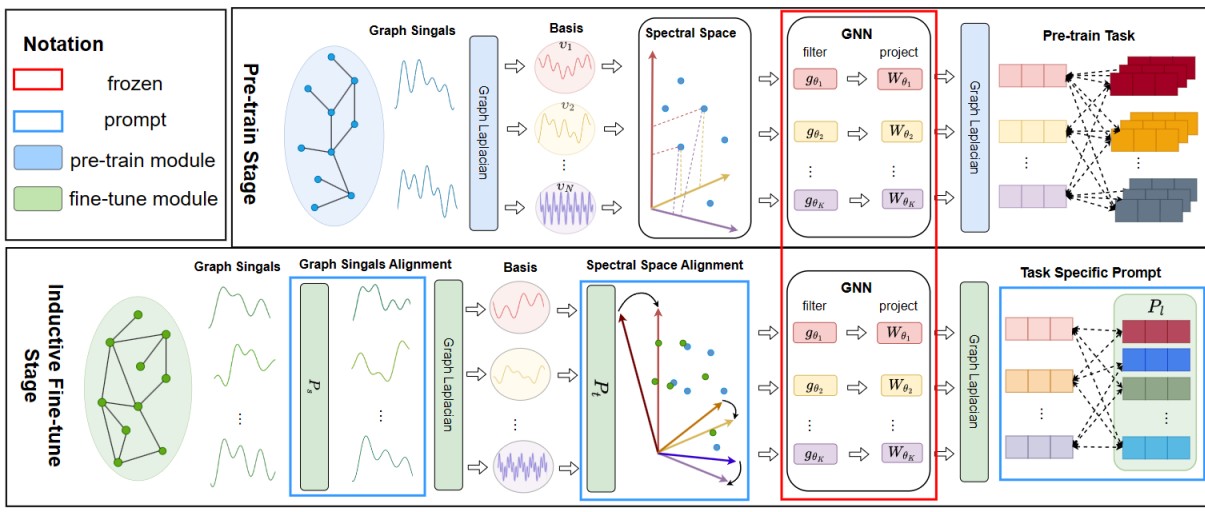

**Figure 1: The framework of IGAP. We first align the graph signals and then we align the spectral space between the pre-train graph and fine-tune graph thus the pre-trained GNN model can be applied. A task-specific prompt is used to align the pre-train task and the fine-tune task.**

where $U = [v_1, v_2, .., v_N]$ is the eigenvectors and $v_i$ is the eigenvector corresponding to the eigenvalue $\lambda_i (\lambda_1 < \lambda_2 < ...\lambda_N)$. The graph signal Fourier transform is defined as:

$$\hat{x} = U^T x \quad \forall x \in X \tag{2}$$

and the inverse graph signal Fourier transform is:

$$x = U\hat{x} \tag{3}$$

**Graph Neural Networks(GNNs).** We use $f_\theta$ to denote the GNN layer parameterized by $\theta$. In spectral domain it can be represented as filter kernel $g_\theta(.)$. The message passing in spectral domain is actually the convolution between the filter kernels and the graph signals, which can be presented as [4]:

$$Z = f_\theta(A, X) = U g_\theta(\Lambda) U^T X = \sum_i v_i g_\theta(\lambda_i) v_i^T x \tag{4}$$

## 4 METHOD

In this section, we first analyze the essence of graph pre-training process from the graph spectral theory. Then based on the analysis, we propose a novel graph prompt method named Inductive Graph Alignment Prompt(IGAP) to deal with the challenges in the inductive fine-tuning, whose framework is shown in Figure 1. At last, we conduct a theoretical analysis to ensure the effectiveness of our proposed method.

### 4.1 Exploring the Essence of Graph Pretraining.

In order to design effective graph prompts for the inductive fine-tuning, it's indispensable to understand the process of graph pre-training. Graph pre-training frameworks can be mainly categorized into three types: (i) link prediction, (ii) subgraph contrastive learning, and (iii) local-global contrastive learning. For the convenience of analysis, we first reformulate these diverse frameworks into an unified one which highlights their common essence.

*4.1.1 Unifying Pre-training Framework.* We contend that all these pre-training frameworks fundamentally operate as a contrastive process, distinguishing positive samples from negative ones. This contrastive form is encapsulated in the InfoNCE loss [13] as follows:

$$InfoNCE = -\sum_i log \frac{sim(\sigma(f_\theta(\mathcal{G}_i)), \sigma(f_\theta(\mathcal{G}_i^+)))}{\sum sim(\sigma(f_\theta(\mathcal{G}_i)), \sigma(f_\theta(\mathcal{G}_i^{-/+})))} \tag{5}$$

where $\mathcal{G}_i$ represents the $i$-th graph view, $\mathcal{G}_i^+$ and $\mathcal{G}_i^-$ are positive and negative samples respectively.[2] $sim(,)$ is the similarity function and $\sigma$ represents a readout head. Different graph pre-training methods are consistent with this formulation, differing only in how they define positive/negative samples and similarity functions.

**Subgraph Contrastive Learning.** This approaches ensure that similar graph views have similar representations, while disparate graph views are distinctly represented [42, 45, 46]. They naturally have the contrastive formation where positive samples are the small perturbated ego-subgraphs centered on the same node and negative samples are the ego-subgraphs centered on different nodes.

**Link Prediction.** The link prediction based methods equip GNNs with universal graph knowledge by predicting randomly masked edges [14]. It increases the link probability between the nodes with masked edges while lowering the probability between the actually nonadjacent nodes. This process can also be represented as the form 5 where positive samples are the subgraphs centered on the nodes with masked edges while negative samples are the subgraphs centered on non-adjacent nodes. The link possibility calculation is denoted as the $sim(,)$ function.

**Local-Global Contrastive Learning.** The local-global contrastive based methods enable GNNs to capture the consistent patterns among the local and global perspectives [27, 36]. These methods assimilate representations between subgraphs and the entire graph

---

[2]$\mathcal{G}_i^{-/+}$ represents all the positive and negative samples.

while maintaining dissimilarity with feature-shuffled graphs. The local-global contrastive based methods also naturally have the InfoNCE formation where the positive samples are the subgraphs while the negative samples are the shuffled ones.

*4.1.2 The Spectral Character of Graph Pre-training.* In order to dig out the essential characteristics of graph pre-training to facilitate the inductive graph fine-tuning, we delve into the pre-training process from a graph spectral perspective. We begin with the conclusion which is presented as the theorem 1, and then provide its proof.

THEOREM 1. *Graph pre-training aligns the graph signal $x$ more with the low-frequent components than the high-frequent components, where $sim(x, v_1) > ... > sim(x, v_N)$ for $\lambda_1 < .. < \lambda_N$.*

For analytical convenience, we use the contrastive pre-training paradigm as an example since we have demonstrated that all the mainstream graph pre-training frameworks can be reformulated as this. There are two major ways to generate positive/negative graph samples: (i) graph structure perturbation and (ii) graph signal perturbation. For the generation of positive samples, the structure perturbation is small and does not rotate the spectral basis, which can be denoted as $v_{i^+} = q_{i^+} + v_i$ where $q_{i^+}$ is small and parallel with $v_i$; The graph signal perturbation actually is a small, angle-stable transformation denoted as a symmetric matrix $F_+ = I + F_{sp}$ where $F_{sp}$ is sparse with small non-zero components. For the generation of negative samples, the structure perturbation introduces high-frequent noise to the spectral basis described as $v_{i^-} = q_{i^-} + v_i$. The graph signal transformation is described as a matrix $F_- = I + F_{dt}$ where $F_{dt}$ is dense. For the graph signal $x$, we have the proof:

PROOF. The message passing process is described as:

$$z = \sum_i v_i g_\theta(\lambda_i) v_i^T x$$
$$z_+ = \sum_i (q_{i^+} + v_i) g_\theta(\lambda_{i^+})(q_{i^+} + v_i)^T (I + F_{sp})x \quad (6)$$
$$z_- = \sum_i (q_{i^-} + v_i) g_\theta(\lambda_{i^-})(q_{i^-} + v_i)^T (I + F_{dt})x$$

Where $z$, $z_+$ and $z_-$ are normalized. The InfoNCE loss can be expressed as:

$$InfoNCE(z, z_+, z_-)$$
$$= -log \frac{\sum_i x^T v_i g_\theta(\lambda_i) g_\theta(\lambda_i^+) v_i^T (I + F_{sp})x + \beta_i}{\sum_{i,j} x^T v_i g_\theta(\lambda_i) v_i^T (q_{j^{+/-}} + v_j) g_\theta(\lambda_{j^{+/-}})(q_{j^{+/-}} + v_j)^T (I + F_{sp/dt})x} \quad (7)$$

Where $\beta_i$ is the influence of $q_{i^+}$ on the graph signal $x$ and is very small. We assume that $q_{j^-} = \sum_i \alpha_{k,j} v_k$ where $\sum_i \alpha_{k,j} = 1$, thus the

formulation 7 is transformed as:

$$InfoNCE(z, z_+, z_-)$$
$$= log\{1 + \frac{\sum_i x^T v_i g_\theta(\lambda_i)(1 + \alpha_{i,i}^2) det(I + F_{dt})_i g_\theta(\lambda_{i^-}) v_i^T x}{\sum_i x^T v_i g_\theta(\lambda_i) det(I + F_{sp})_i g_\theta(\lambda_{i^+}) v_i^T x + \beta_i}$$
$$+ \frac{\sum_{i,j \neq i} x^T v_i g_\theta(\lambda_i)(\alpha_{i,j}) det(I + F_{dt})_j g_\theta(\lambda_{j^-})(\sum_i \alpha_{i,j} v_i + v_j)^T x}{\sum_i x^T v_i g_\theta(\lambda_i) det(I + F_{sp})_i g_\theta(\lambda_{i^+}) v_i^T x + \beta_i}\}$$
$$\approx log\{1 + \frac{\sum_i x^T v_i g_\theta(\lambda_i) \sum_j \alpha_{i,j}(1 + \lambda_{dt,j}) g_\theta(\lambda_{j^-}) v_j^T x}{\sum_i x^T v_i g_\theta(\lambda_i)(1 + \lambda_{sp,i}) g_\theta(\lambda_i^+) v_i^T x} +$$
$$\frac{\sum_i x^T v_i g_\theta(\lambda_i)(1 + \sum_j \alpha_{i,j}^2(1 + \lambda_{dt,j}) g_\theta(\lambda_{j^-})) v_i^T x}{\sum_i x^T v_i g_\theta(\lambda_i)(1 + \lambda_{sp,i}) g_\theta(\lambda_i^+) v_i^T x}\}$$
$$= log\{1 + \frac{\sum_i (v_i^T x) g_\theta(\lambda_i)(1 + \sum_j (\alpha_{i,j} + \alpha_{i,j}^2)(1 + \lambda_{dt,j})) g_\theta(\lambda_{j^-})(v_i^T x)}{\sum_i (v_i^T x) g_\theta(\lambda_i)^2(1 + \lambda_{sp,i})(v_i^T x)}\}$$
$$(8)$$

Where $\lambda_{sp,i}$ is the $i$-smallest the eigenvalue of $F_{sp}$, $\lambda_{dt,i}$ is the $i$-smallest the singular value of $F_{dt}$ and $\lambda_{sp,i} \ll \lambda_{dt,j}$. Since the negative perturbation mainly contains the high-frequent noise corresponding to each spectral components, implying that $\alpha_{i,j} < \alpha_{i+1,j}$ given $\lambda_{i^-} < \lambda_{i+1^-}$. Besides, the graph patterns are also disturbed unevenly with $\lambda_1 < \lambda_{1^-} < ... < \lambda_N < \lambda_{N^-}$. We deduce that $\frac{1 + \sum_j (\alpha_{i,j} + \alpha_{i,j}^2)(1 + \lambda_{dt,j}) g_\theta(\lambda_{j^-})}{g_\theta(\lambda_i)}$ increases as $\lambda_i$ decreases. Therefore, the minimization of loss 8 ensures $v_1^T x > .. > v_N^T x$ given $\sum_i v_i^T x = 1$, thus validating the theorem 1. □

The theorem 1 provides an important clue for inductive fine-tuning: the knowledge of the pre-training process is mainly concentrated on the low-frequent components, which makes it possible to be transferred under the alignment of low-frequent space.

## 4.2 Inductive Graph Alignment Prompt

During the inductive fine-tuning stage, data gap is the most significant challenge which can be attributed to two major sources: (i) graph signal gap and (ii) graph structure gap. The graph signal gap refers to differences in distribution between the fine-tuning graph features and the pre-training ones, while the graph structure gap indicates disparities in their structural properties. In this subsection, we focus on these challenges posed by inductive setting and propose a novel graph prompt based method named Inductive Graph Alignment Prompt(IGAP) to address these gaps.

*4.2.1 Graph Signal Gap.* To address the graph signal gap, we propose a graph signal prompt. We view this gap as a signal perturbation and propose compensating for it by using a learnable graph signal prompt. Specifically, for graph signal $x_i$ the compensation is expressed as follows:

$$\tilde{x}_i = x_i + p_i \quad (9)$$

Where $p_i$ represents the learnable prompt for $x_i$. However, it's expensive if we employ a unique learnable prompt for each signal,

which not only increases the prompt parameters but also raises the risk of over-fitting problem. To mitigate complexity and avoid over-fitting, we propose to utilize a set of graph signal prompts $P_s = [p_{s_1}, .., p_{s_L}]$. The graph signal compensation is then transformed as follows:

$$\tilde{x}_i = x_i + \sum_j \alpha_j^i p_{s_j} \qquad (10)$$

Where $\alpha_j^i$ is also a learnable parameter. By doing so, the complexity, which originally scaled with $O(N \times F)$ for a graph signal matrix of size $N \times F$, is reduced to $O(N \times L + L \times F)$ with $L \ll N$.

### 4.2.2 Graph Structure Gap.
The graph structure gap is essentially the misalignment of spectral space between the pre-training graph and the fine-tuning graph. The pre-trained GNNs cannot be directly applied to the fine-tuning graph because the filters will be invalid in the new space. Fine-tuning GNNs directly might not only compromise the general knowledge but also lead to over-fitting problem. Besides, with the pre-training graph data unavailable, there is no reference for the direct alignment. To fully leverage the pre-trained knowledge, we propose a simple yet effective alignment strategy based on the characteristics of pre-training process. A theoretical analysis is also provided to ensure the effectiveness.

**Spectral Space Alignment: A Recessive Approach** According to the theorem 1, pre-trained GNNs align graph signals more with low-frequent components than high-frequent ones. Consequently, it's possible to align the basis of low-dimensional spectral space corresponding to low-frequent components while maintaining the main knowledge of the pre-trained GNNs. Supposing the pre-training graph and the fine-tuning graph are denoted as $\mathcal{G}_{pt}$ and $\mathcal{G}_{ft}$ respectively. $U_{pt} = [v_{pt_1}, v_{pt_2}, .., v_{pt_N}] \in R^{N \times N}$ and $U_{ft} = [v_{ft_1}, v_{ft_2}, .., v_{ft_M}] \in R^{M \times M}$ are the eigenvectors of $\mathcal{G}_{pt}$ and $\mathcal{G}_{ft}$ respectively. We can reduce the spectral space dimension of the pre-training graph into the fine-tuning one and only consider the alignment of $K$-dimensional spectral subspace based by $U_{pt_K} = [v_{pt_1}, ..., v_{pt_K}] \in R^{M \times K}$ and $U_{ft_K} = [v_{ft_1}, ..., v_{ft_k}] \in R^{M \times K}$. Since $U_{pt_K}$ is inaccessible and we propose a recessive transformation matrix prompt $P_t$ which is learnable:

$$U_{pt_K} = P_t U_{ft_K} \qquad (11)$$

Subsequently, the fine-tuning graph signal can be projected into the aligned spectral space, and the aligned low-frequent signal is calculated as follows:

$$\tilde{Z} = P_t U_{ft_K} g_\theta(\Lambda_{ft}) U_{ft_K}^T P_t^T \tilde{X} \qquad (12)$$

Where the pre-trained knowledged can be transferred recessively to the fine-tuning stage.

**Why Align the $K$-Smallest Spectral Components?** Here we analyze the why the low-frequent spectral components alignment can guarantee the effectiveness under the inductive setting. In the spectral domain, the orthonormal spectral components describe distinct graph smooth patterns and each graph spectral component represents a specific graph smooth pattern. The graph signal encompasses a comprehensive description of all these patterns. Considering a normalized graph signal $x$, we define the informative level and the noise level corresponding to the smooth pattern $i$ as $v_i^T x$ and $1 - v_i^T x$ respectively, since the more compatible between the graph signal and the pattern, the more smooth patterns are

contained, and the less compatible between the graph signal and the pattern, the more noise is contained. The spectral component signal-to-noise ratio is defined as:

$$Sp\_SNR(v_i) = \frac{v_i^T x}{1 - v_i^T x} \qquad (13)$$

The graph signal-to-noise ratio is represented as the average of all the spectral component signal-to-noise ratios:

$$SNR(x) = \frac{1}{N} \sum_i \frac{v_i^T x}{1 - v_i^T x} \qquad (14)$$

This ratio describes the purity of useful patterns in the graph signal. According to Theorem 1, we have $Sp\_SNR(v_1) > Sp\_SNR(v_2) > ... > Sp\_SNR(v_N)$. If we align all the components, it will induce considerable noise. But if we choose less components, many useful patterns will be lost and thus the performance will be compromised. Therefore, we make a balance by aligning the $K$-smallest components.

**Why Does the Spectral Space Have Low Dimension?** According to the theorem 1, the projections of the graph signal on the high-frequent components are smaller, and these different components are mutually orthogonal. Consequently, the information contained in these high-frequent axes in the spectral space is relatively limited. Hence, the spectral space actually has the compacted dimension.

### 4.2.3 Task Type Gap.
During the fine-tuning stage, the task type might also differ from the pre-training one. To preserve the generalization ability of pre-trained GNNs, we propose aligning the task types. We demonstrate that the main kinds of downstream tasks like node classification, graph classification, and link prediction can also be reformulated into the contrastive form. As the link prediction task has been discussed above, here we focus on the node classification task and graph classification task. The Cross-Entropy loss of classification is:

$$CE = \sum_i \sum_j - y_{i,j} log(\sigma_{\theta_j}(z_i)) - (1 - y_{i,j}) log(1 - \sigma_{\theta_j}(z_i)) \qquad (15)$$

Where $y_{i,j}$ is label $j$ of the node $i$, and if we view the parameters of $\sigma_{\theta_j}$ as the label representation $l_j$, the loss in Equation 15 can be transformed into:

$$CE = \sum_i - log \frac{sim(l_i, z_i)}{sim(l_j, z_i)} - log\, sim(l_j, z_i) \qquad (16)$$

The optimization in Equation 16 is equivalent to the optimization of the InfoNCE loss 5. The graph classification task can be transformed in the same way. For classification downstream task with $d$ labels, we formulate the classification InfoNCE loss function as follows using trainable label representations $P_l = [p_1, .., p_d]$:

$$InfoNCE = - \sum_i log \frac{sim(p_i, \tilde{z}_i)}{\sum_j sim(p_j, \tilde{z}_i)} \qquad (17)$$

The inductive fine-tuning optimization problem is expressed as:

$$\underset{(P_s, P_t, P_l)}{argmin} - \sum_i log \frac{sim(p_i, \tilde{z}_i)}{\sum_j sim(p_j, \tilde{z}_i)} \,\forall p_j \in P_l \,\forall \tilde{z}_i \in \tilde{Z} \qquad (18)$$

$$S.T. \quad \tilde{Z} = P_t U_{ft} g_\theta(\Lambda_{ft}) U_{ft}^T P_t^T \tilde{X}$$

Where $g_\theta$ is fixed, $sim(,)$ is the similarity function and in this paper we use cosine similarity.

## 5 EXPERIMENTS

In this section, we conduct experiments on both node classification and graph classification tasks under three distinct settings: (i) transductive setting where the pre-training graph is directly used for fine-tuning; (ii) semi-inductive setting where the pre-training graph and the fine-tuning graph are distinct but share some overlap; (iii) inductive setting where the pre-training graph and the fine-tuning graph have no overlap.

| Datasets | Nodes | Edges | Attributes | Classes |
|---|---|---|---|---|
| Citeseer | 4,230 | 10,674 | 602 | 6 |
| Amazon-Photo | 7,650 | 238,162 | 745 | 8 |
| CoraFull | 19,793 | 126,842 | 8,710 | 70 |
| CoraFull-F | 2,995 | 16,316 | 8,710 | 7 |
| Arxiv-P | 91,605 | 421,382 | 128 | 24 |
| Arxiv-F | 6,337 | 13,364 | 128 | 6 |
| Paper100M-P | 86,428 | 728,614 | 128 | 26 |
| Paper100M-F | 16,892 | 104,732 | 128 | 10 |
| Reddit-P | 51,648 | 2,253,856 | 602 | 18 |
| Reddit-F | 8,680 | 390,616 | 602 | 7 |

**Table 1: Statistics of the node classification datasets**

| Datasets | Graphs | Avg Nodes | Avg Edges | Tasks |
|---|---|---|---|---|
| Molhiv | 41,127 | 25.5 | 54.9 | 1 |
| Molmuv | 93,087 | 24.2 | 52.6 | 17 |
| Moltox21 | 7,831 | 18.6 | 38.6 | 12 |
| Molbace | 1,513 | 34.1 | 73.7 | 1 |
| Molbbbp | 2,039 | 24.1 | 51.9 | 1 |

**Table 2: Statistics of the graph classification datasets**

### 5.1 Datasets and Metrics

The settings of datasets and metrics are described as follows, more details about the data process can be found in Appendix .1:

**Node Classification**: In the transductive setting, we use Citeseer [2] and Amazon-Photo [32] for both pre-training and fine-tuning. In the semi-inductive setting, we conduct experiments on two dataset pairs [2, 37]: (CoraFull, CoraFull-F) and (Arxiv-P, Arxiv-F) where CoraFull and Arxiv-P are used for pre-training while CoraFull-F and Arxiv-F are used for fine-tuning. CoraFull-F is sampled from CoraFull; Arxiv-P and Arxiv-F are sampled from Arxiv; Notably, in the semi-inductive setting, the nodes selected for fine-tuning are also part of the pre-training datasets. In the inductive setting, we conduct experiments on two dataset pairs [12]: (Paper100M-P, Paper100M-F) and (Reddit-P, Reddit-F) [1]. Paper100M-P and Reddit-P are used for pre-training while Paper100M-F and Reddit-F are used for fine-tuning. Paper100M-P and Paper100M-F are sampled from Paper100M; Reddit-P and Reddit-F are sampled from Reddit; In the inductive setting, there is no overlap between nodes and labels. The information on these datasets can be found in Table 1. We sample 100 nodes per class for the training set in Citeseer, Amazon-Photo, and CoraFull-F. For Arxiv-F, Paper100M-F, and Reddit-F, we sample 150, 250, and 250 nodes respectively. For all the fine-tuning datasets the remaining nodes are randomly split as 2:8 for evaluation and testing. We use classification accuracy as metrics. The statistics of these datasets can be found in Table 1.

**Graph Classification**: In the transductive setting, we conduct experiments on Molhiv and Moltox21 [40]. In the semi-inductive setting, we use Molhiv for pre-training and Molbace, Molbbbp for fine-tuning. In the inductive setting, GNNs are pre-trained on Molmuv and fine-tuned on Molbace and Molbbbp as there is no overlap between datasets. For all these datasets, we use RDKit[3] [19] to pre-process them as their official settings and we randomly split these datasets as 4:2:4 for training, evaluation and testing. ROC-AUC is employed as the evaluation metric for all graph classification datasets. The statistics of these datasets can be found in Table 2.

### 5.2 Baselines

To evaluate the effectiveness of our method, we compare it with several state-of-the-art baselines, which can be categorized as follows:

**Supervised Learning**: We train GCN [18], GraphSAGE [12] and GAT [35] from scratch on the fine-tuning graphs in the supervised manner.

**Pre-training+Fine-tuning**: We pre-train GNNs in the pre-training graphs then fine-tune them. The pre-training methods we use are GraphCL [42] and DGI [36].

**Pre-training+Prompt Fine-tuning**: We pre-train the GNNs and fine-tune them using graph prompts to bridge the task gap. The graph prompt methods we use are GPPT [33], GraphPrompt [25], and All-in-One [34].

### 5.3 Experimental Settings.

We present the settings of all these methods, the detail information can be found in Appendix .2.

**Model Settings.** We use 2-layer GNN with 128 hidden neurons as the backbone. For the supervised baselines, we append a 2-layer Neural Network with Relu as the task head. For the graph pre-training baselines, we pre-train GNNs with a head and replace it with a new trainable head for fine-tuning, with the pre-trained GNNs frozen. For the graph prompt baselines, we use the official suggested templates to design prompts on the fine-tuning datasets. We apply grid search to find the optimal model hyperparameters for all these baselines. For IGAP, we set $L$ as 16 and $K$ as 32. We use GraphCL as the pre-training framework and use a new 2-layers Neural Network with ReLU as the task-specific head during fine-tuning. Only the task-specific head and the prompts are trainable with the GNNs frozen. For the graph classification tasks, we use mean pooling to calculate graph representation for all the baselines.

**Training Settings.** For the pre-training and supervised learning baselines, the learning rate is set as 0.0001. The maximum epoch number is set to 500, saving checkpoint every 50 epochs. For the

---

[3]Open-source cheminformatics; http://www.rdkit.org

| Methods | Transductive | | Semi-Inductive | | Inductive | |
|---|---|---|---|---|---|---|
| | Citeseer | Amazon-Photo | CoraFull-F | Arxiv-F | Paper100M-F | Reddit-F |
| GCN | 70.76% | 90.62% | 75.76% | 86.26% | 71.13% | 89.48% |
| GraphSAGE | 71.12% | 89.88% | 76.12% | 86.08% | 71.89% | 88.25% |
| GAT | 69.75% | 90.09% | 75.05% | 86.47% | 72.28% | 88.12% |
| GraphCL+GCN | 73.81% | 91.14% | 77.26% | 84.73% | 67.34% | 85.50% |
| GraphCL+GraphSAGE | 73.56% | 91.61% | 77.56% | 85.05% | 68.92% | 86.33% |
| DGI+GCN | 72.94% | 90.36% | 75.94% | 84.32% | 66.24% | 85.23% |
| DGI+GraphSAGE | 73.69% | 90.45% | 76.69% | 85.56% | 65.32% | 85.55% |
| GPPT+GCN | 73.10% | **92.54%** | 76.10% | 83.71% | 68.88% | 86.04% |
| GPPT+GraphSAGE | 72.53% | 91.89% | 76.53% | 84.80% | 66.98% | 85.61% |
| GraphPrompt+GCN | 74.08% | 92.22% | 77.08% | 84.58% | 68.51% | 85.67% |
| GraphPrompt+GraphSAGE | 74.15% | 92.18% | 78.15% | 83.16% | 70.12% | 84.29% |
| All-in-One+GCN | 73.64% | 92.24% | 77.88% | 85.63% | 66.51% | 85.10% |
| All-in-One+GraphSAGE | **75.11%** | 91.82% | 77.81% | 85.06% | 69.72% | 86.85% |
| IGAP+GCN | 74.44% | 91.84% | **79.46%** | 87.15% | 72.16% | **90.06%** |
| IGAP+GraphSAGE | 74.23% | **92.78%** | 79.23% | 87.69% | 72.74% | 90.35% |
| IGAP+GAT | **74.55%** | 91.35% | 78.55% | **87.55%** | 73.68% | 89.56% |

**Table 3: Performance of node classification task. The best two results are bold.**

| Methods | Transductive | | Semi-Inductive | | Inductive | |
|---|---|---|---|---|---|---|
| | Molhiv | Moltox21 | Molbace | Molbbbp | Molbace | Molbbbp |
| GCN | 0.7606 | 0.7298 | 0.7812 | 0.6523 | 0.7812 | 0.6523 |
| GraphSAGE | 0.7532 | 0.7310 | 0.7895 | 0.6458 | 0.7895 | 0.6458 |
| GraphCL+GCN | 0.7678 | 0.7330 | 0.7991 | 0.6651 | 0.7525 | 0.6389 |
| GraphCL+GraphSAGE | 0.7775 | 0.7418 | 0.7948 | 0.6605 | 0.7677 | 0.6456 |
| GPPT+GCN | 0.7563 | 0.7361 | 0.7763 | 0.6572 | 0.7717 | 0.6434 |
| GPPT+GraphSAGE | 0.7619 | 0.7366 | 0.7956 | 0.6626 | 0.7639 | 0.6466 |
| All-in-One+GCN | **0.7936** | 0.7447 | 0.7950 | 0.6788 | 0.7671 | 0.6550 |
| All-in-One+GraphSAGE | 0.7865 | **0.7529** | 0.7963 | **0.6819** | 0.7740 | 0.6562 |
| IGAP+GCN | **0.7886** | 0.7435 | **0.8041** | 0.6796 | **0.7902** | **0.6644** |
| IGAP+GraphSAGE | 0.7752 | **0.7472** | **0.8022** | **0.6837** | **0.7933** | **0.6629** |

**Table 4: Performance of graph classification task. The best two results are bolded.**

fine-tuning stage, the learning rate is set as 0.001. The maximum epoch number is 100 for all the baselines and we save checkpoint ever 10 epochs. We only report the best performance among all the checkpoints for each baseline. For both stages, we use the Adam [17] without weight decay as the optimizer.

## 5.4 Effectiveness

The node classification and graph classification results are presented in Table 3 and Table 4 respectively. More baselines are tested in Appendix .3. Here are the key observations:

**Node Classification:**

- **Transductive Setting**: Graph pre-training based methods outperform supervised learning methods because pre-training allows GNNs to grasp more universal knowledge and prevent over-fitting problem. Graph prompt based

methods outperform graph pre-training by narrowing down the gap between pre-training and fine-tuning tasks. IGAP achieves competitive performance compared to graph prompt based methods as it also bridges the gap.

- **Semi-inductive Setting**: Graph pre-training and graph prompt based methods outperform supervised methods in some cases(e.g., CoraFull) but not in other(e.g., Arxiv). The possible reason is that CoraFull-F is more compatible with CoraFull but Arxiv-P and Arxiv-F have more gap. Graph prompt based methods bridge the task type gap and thus perform better than the pre-training based methods. Our method achieves the best performance as it succeed in narrowing down the data gap.

- **Inductive Setting**: Graph pre-training based methods perform worse than supervised learning, demonstrating that

| Method | Paper100M-F | Reddit-F |
|---|---|---|
| GraphSAGE | 71.13% | 88.25% |
| GraphCL | 68.92% | 86.33% |
| All-in-One | 69.72% | 86.85% |
| IGAP-GraphSAGE | 72.74% | 90.35% |
| IGAP(No $P_s$) | 70.58% | 88.26% |
| IGAP(No $P_t$) | 68.67% | 87.44% |
| IGAP(No $P_l$, end2end) | 72.01% | 89.28% |

**Table 5: Different Prompt Influence in Inductive Setting.**

| $L$ | Paper100M | Reddit |
|---|---|---|
| 8 | 71.96% | 89.17% |
| 16 | 72.74% | 90.35% |
| 32 | 72.93% | 91.61% |
| 64 | 71.99% | 90.88% |

**Table 6: Influence of $L$.**

| $K$ | Paper100M | Reddit |
|---|---|---|
| 16 | 71.24% | 89.46% |
| 32 | 72.74% | 90.35% |
| 64 | 73.22% | 90.59% |
| 128 | 73.32% | 90.25% |

**Table 7: Influence of $K$.**

the data gap will result in negative transfer to the downstream tasks. Graph prompt based methods also suffer significantly because they fail to bridge the data gap. Our method outperforms all the baselines, demonstrating its success in bridging the data gap in the inductive setting.

**Graph Classification:**

- **Transductive and Semi-inductive Settings**: Graph pre-training based methods outperform supervised learning, indicating that the general pre-trained knowledge can mitigate the over-fitting problem. Graph prompt-based methods narrowing down the task type gap and thus have some improvements. Our proposed method achieves better performance compared with other graph prompt based methods.

- **Inductive Setting**: Graph pre-training and graph prompt-based methods do not perform well because the pre-trained knowledge cannot be directly applied to dissimilar fine-tuning graphs. It's worthy to note that there is less negative effect compared to the node classification task in the inductive setting, the main reason lies in: the molecular graphs have relative simple graph patterns thus the knowledge are easy to be transferred. Our proposed method achieves better performance, illustrating the effectiveness of narrowing down the data gap.

## 5.5 Ablation Study

To demonstrate the effectiveness of different prompt modules, we conduct an ablation study in inductive node classification task and the results are shown in Table 5, the graph classification results can be found in Appendix .4. From the results, we find that the performance significantly decreases without spectral space alignment, indicating the crucial role of the alignment in inductive scenario. Besides, no signal alignment also results in a decrease in performance but is not as much as no space alignment, the possible reason might be that the space alignment can compensate the graph signal gap. At last, fine-tuning end-to-end has a minor impact on performance. This might be attributed to that the general knowledge can compensate for the task type gap.

## 5.6 Hyperparameter Study

We conduct experiments to test the influence of hyperparameters on node classification task and more results can be found in Appendix .5. We set the $L$ as 8, 16, 32, 64 and $K$ as 16, 32, 64, 128 and the results can be found in Table 6 and Table 7 respectively. We find small $L$

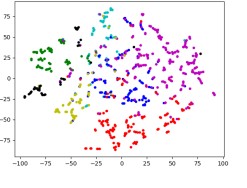

(a) Visualization of GraphSage.

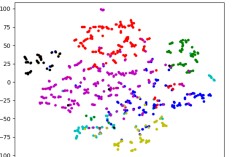

(b) Visualization of IGAP.

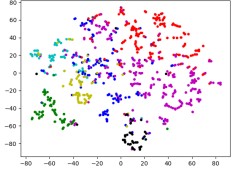

(c) Visualization of GraphPrompt.

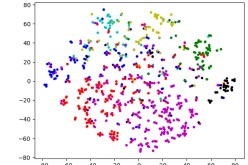

(d) Visualization of GraphCL.

**Figure 2: Visualization of Different Baselines on Reddit-F.**

and $K$ will make the alignment difficult because of less parameters but if we set $L$ and $K$ too large it will result in higher costs and make it hard for fine-tuning with redundant parameters.

## 5.7 Visualization

We randomly sample 300 nodes from Reddit-F and visualize the representations of different baselines by t-SNE, the results are shown in Figure 2. GraphCL and GraphPrompt have difficult in discriminating some node classes because the pre-trained GNNs have compromised performance in inductive scenario. IGAP has better cluster property which corroborates the effectiveness of the alignment.

## 6 CONCLUSION

In this paper, we propose a novel graph prompt based method to deal with the data gap in inductive fine-tuning. We first analyze the essence of graph pre-training process under an unified framework. Then for the inductive fine-tuning stage, we identify the two main sources of the data gap: (i) graph signal gap and (ii) graph structure gap. Based on the insight of graph pre-training, we propose to align the graph signal and spectral space with the learnable prompts. Theoretical analysis is also given to justify our method. Extensive experiments shows the effectiveness in bridging the data gap under the inductive setting.

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

## .1 Data Process

In this subsection we dilate the data process under different experimental settings: (i) transductive setting, (ii) semi-inductive settings and (iii) inductive settings. We conduct node classification and graph classification tasks respectively.

| Datasets | Nodes | Edges | Attributes | Classes |
|---|---|---|---|---|
| Citeseer | 4,230 | 10,674 | 602 | 6 |
| Amazon-Photo | 7,650 | 238,162 | 745 | 8 |
| CoraFull | 19,793 | 126,842 | 8,710 | 70 |
| CoraFull-F | 2,995 | 16,316 | 8,710 | 7 |
| Arxiv-P | 91,605 | 421,382 | 128 | 24 |
| Arxiv-F | 6,337 | 13,364 | 128 | 6 |
| Paper100M-P | 86,428 | 728,614 | 128 | 26 |
| Paper100M-F | 16,892 | 104,732 | 128 | 10 |
| Reddit-P | 51,648 | 2,253,856 | 602 | 18 |
| Reddit-F | 8,680 | 390,616 | 602 | 7 |

**Table 8: Statistics of the node classification datasets**

The datasets process for node classification tasks can be described as follows, whose statistic information can be found in Table 8.

- **Transductive Setting:** We use Citeseer [2] and Amazon-Photo [32] for the transductive experiments, where the same dataset is used for pre-training and fine-tuning. In the fine-tuning stage, we randomly sample 100 nodes per class as the training set, and the remaining nodes are randomly split as 2:8 for training evaluation and testing.
- **Semi-Inductive Setting:** We conduct graph pre-training on two kinds of dataset: CoraFull and Arxiv-P [2, 37], then we fine-tune the pre-trained models on the subgraphs corresponding to these two datasets: CoraFull-F and Arxiv-F. Arxiv-P is randomly sampled from Arxiv which contains the 24 node classes. Then for the fine-tuning subgraphs, we randomly sample a part of classes in the pre-training dataset and the subgraphs specifc to these classes are used for fine-tuning. The class numbers of CoraFull-F and Arxiv-F are 7 and 6 respectively. We randomly sample 100 and 150 nodes per class for the CoraFull-F and Arxiv-F fine-tuning respectively. The remaining nodes are randomly split as 2:8 for training evaluation and testing.
- **Inductive Setting:** We conduct graph pre-training on two kinds of dataset: Paper100M-P [12] and Reddit-P [1], then we fine-tune the pre-trained models on another two datasets: Paper100M-F and Reddit-F. For the pre-training datasets, Paper100M-P and Reddit-P are sampled from Paper100M and Reddit based on the randomly selected classes. The class numbers of Paper100M-P and Reddit-P are 26 and 18 respectively. As for the fine-tuning datasets, we randomly sample another part of classes in the Paper100M and Reddit without overlap with the pre-training classes. The graphs specific to these classes are used for fine-tuning. The class numbers of Paper100M-F and Reddit-F are 10 and 7 respectively. We randomly sample 250 nodes per class for both the Paper100M-F and Reddit-F fine-tuning. The remaining nodes are randomly split as 2:8 for training evaluation and testing.

| Datasets | Graphs | Avg Nodes | Avg Edges | Tasks |
|---|---|---|---|---|
| Molhiv | 41,127 | 25.5 | 54.9 | 1 |
| Molmuv | 93,087 | 24.2 | 52.6 | 17 |
| Moltox21 | 7,831 | 18.6 | 38.6 | 12 |
| Molbace | 1,513 | 34.1 | 73.7 | 1 |
| Molbbbp | 2,039 | 24.1 | 51.9 | 1 |

**Table 9: Statistics of the graph classification datasets**

The datasets process for graph classification tasks can be described as follows, whose statistic information can be found in Table 9.

- **Transductive Setting:** We conduct experiments Molhiv and Moltox21 [40], where the same dataset is used for pre-training and fine-tuning. In the fine-tuning stage, we randomly split these datasets as 4:2:4 for training, evaluation and testing. RDKit[4] [19] is used to pre-process them as their official settings.
- **Semi-Inductive Setting:** We conduct graph pre-training on Molhiv and the fine-tune the pre-trained GNNs on Molbace and Molbbbp. There is overleap between the pre-training dataset and the fine-tuning ones because Molbace and Molbbbp are the sub-datasets of Molhiv. In the fine-tuning stage, we split Molbace and Molbbbp as 4:2:4 for training, evaluation and testing and we also use RDKit to pre-process them.
- **Inductive Setting:** We conduct graph pre-training on Molmuv and then we fine-tune the pre-trained GNNs on Molbace and Molbbbp. There is no overlap between the pre-training dataset and the fine-tuning ones. The process of the fine-tuning datasets is the same as the semi-inductive setting.

## .2 Experimental settings

In this subsection we describe the experimental settings in detail. **Baseline Settings.** We use 2-layer GNN with 128 hidden dimensions for all the baselines as the backbone model. For the supervised learning based methods [12, 18, 35] we use a 2-layer Neural Network with Relu as the task-specific head. For the graph pre-training based methods, we have the following settings: (i) For GraphCL [42] we use feature perturbation, edge mask and subgraph mask as the data augmentation. (ii) For DGI [36] we use the mean-pooling as the readout function and 2-layer Neural Network with LeakRelu as the representation head. Binary-Cross-Entropy loss is used as the pre-training loss function. Signal shuffle is used as the negative samples generation function as the paper suggests. After the pre-training stage, the GNNs are frozen, and we use a new task-specific head which is a 2-layer Neural Network for fine-tuning. For the graph prompt based methods, we have the following settings: (i) For GPPT [33] we adapt the official settings using learnable task tokens

---

[4]Open-source cheminformatics; http://www.rdkit.org

and structure tokens. Since in the semi-inductive and inductive settings the pre-training graphs are unavailable, we use the training nodes from fine-tuning graphs to construct the structure tokens. (ii) For GraphPrompt [25] we use the learnable readout function to reformulate the node classification task to the pre-training form as the papper suggests, for the graph classification task we use the mean-pooling as the graph representation function and reformulate in the same way. (iii) For All-in-One [34], we reformulate the node classification task and graph classification task into graph-level task as the paper does. The prompt tokens and structure tokens are designed in the same way as the paper on the inductive fine-tuning datasets, we also use the meta-learning to initialize the prompt parameters. We apply grid search to find the optimal model hyperparameters for each baseline. For IGAP, we set $L$ as 16 and $K$ as 32 as default. We use GraphCL as the pre-training framework whose hyperparameters are decided through grid search. In the inductive fine-tuning, we a new 2-layer Neural Network as the task specific head for fine-tuning. We only train the head parameters and the prompt parameters with the parameters of GNN frozen. For all the baselines, we use mean-pooling to calculate the graph representation for the graph classification tasks.

**Experimental Settings.** For the pre-training and supervised learning baselines, the learning rate is set as 0.0001 and the maximum training epoch number is set as 500. We save checkpoint every 50 epochs for the supervised learning baselines and pre-training methods. The best performance of each supervised learning baseline is used. For the graph fine-tuning and graph prompt fine-tuning, we set the learning rate as 0.001 and we fine-tune the 100 epochs with saving checkpoint every 10 epochs. We test the last 4 pre-training checkpoints with all the 10 fine-tuning checkpoints for every baseline(40 results in total for every method) and we only report the best performance. For both stages we use Adam [17] without weight decay as the optimizer.

## .3  More Baselines Experiments

In this subsection we conduct node classification tasks and graph classification tasks on more baselines, the baselines are descirbed as:

**Supervised Learning**: We train GCN [18], GraphSAGE [12], GAT [35] and GIN [41] from scratch on the fine-tuning graphs in a supervised manner.

**Pre-training+Fine-tuning**: We pre-train GNNs in the pre-training graphs then fine-tune them. The pre-training methods we use are GraphCL [42], DGI [36], GCA [46] and GPT-GNN[14].

**Pre-training+Prompt Fine-tuning**: We pre-train the GNNs and fine-tune them using graph prompts to bridge the task gap. The graph prompt methods we use are GPPT [33], GraphPrompt [25], and All-in-One [34].

**Graph Transfer Learning:** Although graph transfer learning is not fit for the inductive setting, we provide the pre-training datasets in the fine-tuning stage to testify some sate-of-the-art graph transfer learning baselines. The baselines we use include: DANE[44], UDA-GCN [39]. Notably, these methods are proposed for node representation alignment thus not fit for the graph classification tasks.

As for the newly added baselines, the layer number of GIN is set as 2 and the 2-layer Neural Network is used as the reduce function. The training setting is the same as the GCN. As for GCA and GPT-GNN, the hyperparameters are also decided by grid search and the pre-training setting is the same as GraphCL. For DANE and UDA-GCN, we provide the pre-training datasets for their alignment. We use their official training strategies and search the best hyperparameters. We pre-train 500 epochs and fine-tune 300 epochs for DANE and UDA-GCN. We save the model checkpoint ever 50 epochs, we report the best performance among all the checkpoints. The experimental results are shown in Table 10 and Table 11.

**Node Classification Task.** For the node classification task, we have the following observations:

- **Transductive setting:** As the observation in the experimental results, graph pre-training based methods and graph prompt based methods perform better than the supervised learning based methods, which means the general knowledge can mitigate over-fitting problem. Transfer learning based methods do not perform as well as the pre-training based methods, the possible reason lies that the domain adaptation hurts some general knowledge. Our methods perform achieves comparable performance with the prompt based methods.

- **Semi-Inductive setting:** Graph pre-training based methods perform better than the supervised learning based methods on CoraFull-F mainly because the fine-tuning dataset has less gap with the pre-training one. But the performance will deteriorate as the gap increase(e.g. Arxiv-F). GPT-GNN performs worse than other pre-training baselines, it's because GPT-GNN is more sensitive to the structural perturbation. Graph prompt based methods achieve the improved performance compared to the pre-training baselines mainly because they narrow down the task type gap. Transfer learning based methods perform better than the prompt based methods because they make use of the patterns of pre-training graphs to align some of the representations. Our proposed method achieves the best performance, showing the success in bridging the data gap.

- **Inductive setting:** Graph pre-training based methods and graph prompt based methods perform worse than the supervised learning, which demonstrates that the data gap will lead to negative transfer. The transfer learning based methods achieve comparable performance with the supervised methods because they make use of the pre-training graph and revise the unmatched graph patterns, but hurts the generalization ability. Our proposed method achieves better performance than all the baselines which shows the effectiveness of narrowing down the data gap.

**Graph Classification Task.** For the graph classification task, we have the following observations:

- **Transductive setting And Semi-Inductive Settings:** Graph pre-training based methods and graph prompt base methods perform better than the supervised baselines because the pre-trained knowledge improve the downstream tasks and avoid over-fitting when fine-tuning. However, GPT-GNN perform not as well as other pre-training baselines,

| Methods | Transductive | | Semi-Inductive | | Inductive | |
|---|---|---|---|---|---|---|
| | Citeseer | Amazon-Photo | CoraFull-F | Arxiv-F | Paper100M-F | Reddit-F |
| GCN | 70.76% | 90.62% | 75.76% | 86.26% | 71.13% | 89.48% |
| GraphSAGE | 71.12% | 89.88% | 76.12% | 86.08% | 71.89% | 88.25% |
| GAT | 69.75% | 90.09% | 75.05% | 86.47% | 72.28% | 88.12% |
| GIN | 71.59% | 90.56% | 75.03% | 86.13% | 71.24% | 89.20% |
| GraphCL+GCN | 73.81% | 91.14% | 77.26% | 84.73% | 67.34% | 85.50% |
| GraphCL+GraphSAGE | 73.56% | 91.61% | 77.56% | 85.05% | 68.92% | 86.33% |
| DGI+GCN | 72.94% | 90.36% | 75.94% | 84.32% | 66.24% | 85.23% |
| DGI+GraphSAGE | 73.69% | 90.45% | 76.69% | 85.56% | 65.32% | 85.55% |
| GCA+GCN | 72.46% | 90.34% | 76.70% | 84.33% | 68.96% | 86.58% |
| GCA+GraphSAGE | 73.19% | 91.33% | 77.32% | 84.82% | 68.56% | 86.07% |
| GPT-GCN | 72.50% | 89.56% | 74.11% | 81.89% | 63.41% | 83.34% |
| GPT-GraphSAGE | 71.61% | 90.51% | 74.78% | 82.62% | 62.42% | 83.25% |
| GPPT+GCN | 73.10% | **92.54%** | 76.10% | 83.71% | 68.88% | 86.04% |
| GPPT+GraphSAGE | 72.53% | 91.89% | 76.53% | 84.80% | 66.98% | 85.61% |
| GraphPrompt+GCN | 74.08% | 92.22% | 77.08% | 84.58% | 68.51% | 85.67% |
| GraphPrompt+GraphSAGE | 74.15% | 92.18% | 78.15% | 83.16% | 70.12% | 84.29% |
| All-in-One+GCN | 73.64% | 92.24% | 77.88% | 85.63% | 66.51% | 85.10% |
| All-in-One+GraphSAGE | **75.11%** | 91.82% | 77.81% | 85.06% | 69.72% | 86.85% |
| DANE+GCN | 72.02% | 90.33% | 77.81% | 85.21% | 88.96% | 87.45% |
| DANE+GraphSAGE | 72.22% | 90.17% | 77.37% | 85.35% | 89.68% | 88.10% |
| UDA-GCN | 71.80% | 89.67% | 77.65% | 85.83% | 71.27% | 89.61% |
| UDA-GraphSAGE | 71.55% | 89.14% | 77.49% | 85.30% | 72.15% | 89.28% |
| IGAP+GCN | 74.44% | 91.84% | **79.46%** | 87.15% | 72.16% | **90.06%** |
| IGAP+GraphSAGE | 74.23% | **92.78%** | 79.23% | 87.69% | 72.74% | **90.35%** |
| IGAP+GAT | **74.55%** | 91.35% | 78.55% | **87.55%** | **73.68%** | 89.56% |

Table 10: Performance of node classification task. The best two results are bold.

the possible reason is that GPT-GNN relies too much on the graph structure information while the molecular graphs have relative monotonous graph patterns compared to the large social networks. Our method can achieve competitive and even better performance compared with the graph prompt based methods.

- **Inductive setting:** Graph pre-training based methods and graph prompt based methods perform not as well as the supervised learning methods because the disparate graph patterns might invalidate some of the pre-training knowledge. GPT-GNN perform much worse for the same reason mentioned before. Our proposed method achieves better performance as it mitigates the influence of data gap.

## .4 Ablation Study For Inductive Graph Classification

In this subsection we conduct ablation study on inductive graph classification tasks, the results are shown in Table 12. From the results we find the similar observations with the node classification tasks. The performance will decrease a lot without spectral space and graph signal alignment but it has less influence when no label prompt, the reason might lie in that the transferred knowledge compensate the influence of task type. Particularly, we find the

influence of no spectral space alignment is not as significant as the node classification tasks, it mainly because the graph patterns in molecular graph are more compatible among different datasets as the graph is relative small.

## .5 Hyperparameter Study For Inductive Graph Classification

In this subsection we conduct hyperparameter study on inductive graph classification tasks, the results are shown in Table 13 and Table 14 respectively. From the results we find that the small $L$ will have negative influence on the performance of inductive graph fine-tuning but large $L$ have limited improvement, which is consistent with the observations of the node classification tasks. Besides, $K$ has less influence on graph classification tasks which mainly because the molecular graphs have relative compatible graph patterns.

| Methods | Transductive | | Semi-Inductive | | Inductive | |
|---|---|---|---|---|---|---|
| | Molhiv | Moltox21 | Molbace | Molbbbp | Molbace | Molbbbp |
| GCN | 0.7606 | 0.7298 | 0.7812 | 0.6523 | 0.7812 | 0.6523 |
| GraphSAGE | 0.7532 | 0.7310 | 0.7895 | 0.6458 | 0.7895 | 0.6458 |
| GAT | 0.7523 | 0.7160 | 0.7891 | 0.6439 | 0.7747 | 0.6536 |
| GIN | 0.7561 | 0.7289 | 0.7844 | 0.6426 | 0.7886 | 0.6582 |
| GraphCL+GCN | 0.7678 | 0.7330 | 0.7991 | 0.6651 | 0.7525 | 0.6389 |
| GraphCL+GraphSAGE | 0.7775 | 0.7418 | 0.7948 | 0.6605 | 0.7677 | 0.6456 |
| DGI+GCN | 0.7517 | 0.7276 | 0.7764 | 0.6495 | 0.7331 | 0.6119 |
| DGI+GraphSAGE | 0.7579 | 0.7258 | 0.7627 | 0.6508 | 0.7315 | 0.6077 |
| GCA+GCN | 0.7618 | 0.7367 | 0.7805 | 0.6544 | 0.7452 | 0.6280 |
| GCA+GraphSAGE | 0.7532 | 0.7374 | 0.7854 | 0.6571 | 0.7489 | 0.6325 |
| GPT+GCN | 0.7410 | 0.7205 | 0.7509 | 0.6319 | 0.7211 | 0.5975 |
| GPT+GraphSAGE | 0.7345 | 0.7217 | 0.7646 | 0.6261 | 0.7150 | 0.6019 |
| GPPT+GCN | 0.7563 | 0.7361 | 0.7763 | 0.6572 | 0.7717 | 0.6434 |
| GPPT+GraphSAGE | 0.7619 | 0.7366 | 0.7956 | 0.6626 | 0.7639 | 0.6466 |
| GraphPrompt+GCN | 0.7654 | 0.7389 | 0.7755 | 0.6637 | 0.7603 | 0.6339 |
| GraphPrompt+GraphSAGE | 0.7767 | 0.7429 | 0.7861 | 0.6610 | 0.7615 | 0.6396 |
| All-in-One+GCN | **0.7936** | 0.7447 | 0.7950 | 0.6788 | 0.7671 | 0.6550 |
| All-in-One+GraphSAGE | 0.7865 | **0.7529** | 0.7963 | **0.6819** | 0.7740 | 0.6562 |
| IGAP+GCN | **0.7886** | 0.7435 | **0.8041** | 0.6796 | **0.7902** | **0.6644** |
| IGAP+GraphSAGE | 0.7752 | **0.7472** | **0.8022** | **0.6837** | **0.7933** | **0.6629** |

**Table 11: Performance of graph classification task. The best two results are bolded.**

| Method | Molbace | Molbbbp |
|---|---|---|
| GraphSAGE | 0.7895 | 0.6458 |
| GraphCL | 0.7677 | 0.6456 |
| All-in-One | 0.7740 | 0.6562 |
| IGAP-GraphSAGE | 0.7933 | 0.6629 |
| IGAP(No $P_s$) | 0.7732 | 0.6409 |
| IGAP(No $P_t$) | 0.7696 | 0.6445 |
| IGAP(No $P_l$, end2end) | 0.7812 | 0.6615 |

**Table 12: Different Prompt Influence in Inductive Setting On Graph Classification.**

| $L$ | Molbace | Molbbbp |
|---|---|---|
| 8 | 0.7876 | 0.6556 |
| 16 | 0.7933 | 0.6629 |
| 32 | 0.7912 | 0.6698 |
| 64 | 0.7945 | 0.6631 |

**Table 13: Influence of $L$.**

| $K$ | Molbace | Molbbbp |
|---|---|---|
| 16 | 0.7915 | 0.6590 |
| 32 | 0.7933 | 0.6629 |
| 64 | 0.7926 | 0.6638 |
| 128 | 0.7957 | 0.6634 |

**Table 14: Influence of $K$.**

