# OpenReview forum: "Inductive Graph Alignment Prompt: Bridging the Gap between Graph Pre-training and Inductive Fine-tuning From Spectral Perspective."
_ACM.org/TheWebConf/2024/Conference — TheWebConf24 Oral_

### Official Review · Reviewer_eHiA · 2023-11-20

**Novelty:** 5
**Technical Quality:** 4

**Review:**

This paper propose a graph prompting technique that improves the fine-tuning performance when the fine-tuning graph structures and features are not similar with pre-training graphs. Specifically, it adds two different prompts to minimize the structural and feature divergence. The latter one is motivated by one theoretical results on the importance of low-frequent components on spectral. The experiments demonstrate the effectiveness on transduction, semi-inductive and inductive settings.

I think the storyline of paper is very clear supported by several key steps: (1) analyze the "effects" of pre-training on fine-tuning graphs (2) unify three representative tasks on graph structured data (3) two prompts on structural and feature divergence respectively, while structural divergence is supported by (1).

Nevertheless, there are places that still confuses me to support this paper's acceptance. At high level:
1. Design on "graph structural" gap do not have theoretical support nor experimental evidence.
2. Experiments set-ups seem not following standard practice.
See question sections for more details.

**Questions:**

1. As the author mentioned, the Eq 11 can not access the spectral of pre-training graphs. Therefore, based on my understanding, the design of graph structural prompt is based on "intuitive" instead of "theoretical" observation. In order to practically proves its effectiveness, I suggest the author to provide a synthetic result when $U_{pt_k}$ is known and replacing $P_t U_{ft_K}$ in Eq.18. I am curious to see whether the framework still works and if it works whether the model performance is better than the learned prompt. I believe it can help me understand the function of $P_t$.

2. All experimental results do not have standard deviations? Do authors conducted repeated experiments here?

3. Specifically, I don't understand why the model comparison is conducted by picking "best checkpoint" on test data. I believe the common practice is to use the validation on fine-tuning graph to select **one** checkpoint for every baseline including IGAP.

**Reviewer Confidence:**

3: The reviewer is confident but not certain that the evaluation is correct

**Scope:**

3: The work is somewhat relevant to the Web and to the track, and is of narrow interest to a sub-community

---

### Official Review · Reviewer_nMLh · 2023-11-22

**Novelty:** 5
**Technical Quality:** 5

**Review:**

The paper proposes Inductive Graph Alignment Prompt addressing the data gap between graph pre-training and inductive fine-tuning. Leveraging insights from graph spectral theory, the authors introduce learnable prompts to align graph signals and structures, showing its effectiveness in various settings.

Pros:
- IGAP introduces a unique approach by combining graph pre-training with inductive fine-tuning, addressing the significant gap between various graph data scenarios and the authors provide a solid theoretical foundation, grounded in graph spectral theory. The authors conduct extensive experiments across various settings (transductive, semi-inductive, and inductive) showcase the versatility of IGAP, indicating its ability to adapt to different scenarios effectively.

Cons:
- The proposal of learnable prompts introduces complexity to the model, and the paper should address concerns related to potential overfitting and increased computational requirements. The authors assume compatibility between pre-training and fine-tuning graphs, which might not hold true in real-world scenarios.

**Questions:**

Would like if the authors can propose a robustness study to analyze if IGAP can maintain its effectiveness when faced with noisy or incomplete pre-training data (considering the inaccessibility of the pre-training dataset)

**Reviewer Confidence:**

4: The reviewer is certain that the evaluation is correct and very familiar with the relevant literature

**Scope:**

4: The work is relevant to the Web and to the track, and is of broad interest to the community

---

### Official Review · Reviewer_EFpj · 2023-11-24

**Novelty:** 6
**Technical Quality:** 6

**Review:**

This paper introduces a novel graph prompt based graph pre-training and fine-tuning method called Inductive Graph Alignment Prompt (IGAP). Through the analysis about the spectral characters of graph pre-training methods, Theorem 1 is proposed that graph pre-training predominantly aligns the graph signal with low-frequent components rather than high-frequent ones, which is interesting. Then based on the graph signal gap, graph structure gap, and task type gap between pre-training and fine-tuning stage, authors propose three different types of prompt to achieve alignments. Experiments show that IGAP combined with different GNNs is effective on transductive, semi-inductive, and inductive setting, and the three types of prompts all contribute to the good performance.

**Questions:**

Why you choose 2-layer GNN with 124 hidden neurons as the backbone? Will the number of layers and dimensions affect the performance?

**Reviewer Confidence:**

3: The reviewer is confident but not certain that the evaluation is correct

**Scope:**

4: The work is relevant to the Web and to the track, and is of broad interest to the community

---

### Official Review · Reviewer_QLFh · 2023-11-24

**Novelty:** 5
**Technical Quality:** 4

**Review:**

This work introduces the "Inductive Graph Alignment Prompt" (IGAP), a novel method addressing the challenges in graph neural networks (GNNs) specifically the gap between pre-training and fine-tuning stages. Pre-training of GNNs improves performance by capturing general knowledge without manual annotations. However, existing methods do not adequately bridge the gap in graph data types between these stages. IGAP aims to resolve this by analyzing the essence of graph pre-training from a spectral theory perspective and addressing two identified gaps: the graph signal gap and the graph structure gap. It proposes using learnable prompts in the spectral space to bridge these gaps, supported by extensive experiments under various settings.

Pros

1. **Innovative Approach:** IGAP is an innovative approach that unifies mainstream graph pre-training frameworks and provides a novel solution to bridge the data gap in GNNs.
2. **Comprehensive Analysis:** The paper thoroughly analyzes the essence of graph pre-training and identifies key gaps in the inductive setting, enhancing the understanding of graph neural networks.
3. **Empirical Validation:** The method is empirically validated with extensive experiments, demonstrating its effectiveness across different settings including transductive, semi-inductive, and inductive scenarios.

Cons
1. **Complexity for Practitioners:** The approach, while thorough, might be complex for practitioners to implement, given its in-depth theoretical underpinnings and reliance on spectral graph theory.
2. **Limited Focus on Scalability:** There is a limited discussion on the scalability of the method when applied to very large graphs, which is a common challenge in real-world applications.
3. **Potential Overfitting Risks:** While the method addresses the generalization gap, there could be risks of overfitting, especially in scenarios with significant differences between pre-training and fine-tuning data. This aspect could be further explored.
4. The target problem should be well formulated in the front

**Questions:**

see in Cons in the above section

**Reviewer Confidence:**

3: The reviewer is confident but not certain that the evaluation is correct

**Scope:**

3: The work is somewhat relevant to the Web and to the track, and is of narrow interest to a sub-community

---

### Official Review · Reviewer_2vCS · 2023-12-03

**Novelty:** 5
**Technical Quality:** 5

**Review:**

This study proposes a model that pre-trains a graph neural network on inductive scenarios. Specifically, to bridge the gap between the pretraining graph and the target graph signal, the model utilizes graph prompt tuning with learnable parameters for each graph signal. Additionally, to reduce the graph structure gap, the graphs are aligned in a low-dimensional spectral space. The authors demonstrate the effectiveness of the proposed method in various transferring experiment settings.

Pros:
- The proposed model addresses a crucial issue of extending beyond transductive scenarios to inductive situations, which many existing prompt-based graph pre-training methods focus on.
- The design of the proposed model is mathematically grounded.
- It proves effective in various pretraining then  fine-tuning scenarios.

Cons:
- It would be beneficial to have repeated experiments and corresponding statistical analyses to eliminate the randomness in the experimental results, especially given some experimental results where the performance gain seems marginal.

Minor:
- For clarity, it's conventional to highlight the best score in bold and the runner-up score with underline when presenting performance scores. Using two bolds might affect readability.
- Improving the clarity of Figure 2 would be advantageous.

**Questions:**

N/A

**Reviewer Confidence:**

2: The reviewer is willing to defend the evaluation, but it is likely that the reviewer did not understand parts of the paper

**Scope:**

4: The work is relevant to the Web and to the track, and is of broad interest to the community

---

### Decision · Program_Chairs · 2024-01-22

**Decision:**

Accept (Oral)

**Comment:**

The paper proposes a novel inductive graph alignment prompt technique that allows generalization to unseen graphs. The authors have performed analysis in the graph spectral domain.

 + Strong analysis
 + Novel method for an overlooked problem with a large potential impact.

 - Limited analysis on scalability
 - Learnable prompts may incur high complexity in the model that should be well characterized.